# Photoredox cooperative N-heterocyclic carbene/palladium-catalysed alkylacylation of alkenes

You-Feng Han[1,2,3], Ying Huang[1,2,3], Hao Liu[1,2], Zhong-Hua Gao[1,2], Chun-Lin Zhang [1] ✉ & Song Ye [1,2] ✉

Three-component carboacylation of simple alkenes with readily available reagents is challenging. Transition metal-catalysed intermolecular carboacylation works for alkenes with strained ring or directing groups. Herein, we develop a photoredox cooperative N-heterocyclic carbene/Pd-catalysed alkylacylation of simple alkenes with aldehydes and unactivated alkyl halides to provide ketones in good yields. This multicomponent coupling reaction features a wide scope of alkenes, broad functional group compatibility and free of exogenous photosensitizer or external reductant. In addition, a series of chlorinated cyclopropanes with one or two vicinal quaternary carbons is obtained when chloroform or carbon tetrachloride is used as the alkyl halide. The reaction involves the alkyl radicals from halides and the ketyl radicals from aldehydes under photoredox cooperative N-heterocyclic carbene/Pd catalysis.

Transition metal (TM)-catalyzed vicinal dicarbofunctionalization of alkenes is a powerful protocol for the construction of two C–C bonds to give value-added molecules from simple starting materials[1–4]. Particularly, the TM-catalyzed alkene carboacylation reactions provide easy access to ketones, which are key structural motifs in many natural products[5,6]. The intramolecular carboacylation under TM catalysis has been well established, including carboacylation with acylquinolines[7–10], reactions of alkenes tethered with ring-strained cyclobutanones[11–13], unstrained ketones[14], and amides[15], etc. However, the intermolecular TM-catalyzed carboacylation has been far less developed. The Rh-catalyzed intermolecular carboacylation of alkenes with the directing group was reported by Douglas in 2009[16]. Later, the Pd- or Ni-catalyzed carboacylation of norbornene with amides or esters using triarylborane as the arylation reagent was developed[17,18]. The nickel-catalyzed carboacyaction with acyl chloride and perfluoroalkyl iodide was recently reported by Chu et al.[19]. Apparently, the requirement of alkenes with a strained ring or directing group and the employment of perfluoroalkyl reagents limits the potential application of those reactions.

N-Heterocyclic carbene (NHC) catalysis has evolved as a powerful tool for the construction of various organic molecules[20–24]. Recently, the NHC-catalyzed radical reactions provide possibility for chemical transformations (Fig. 1a)[25–28]. In 2019, Ishii et al. disclosed the NHC-catalyzed radical decarboxylative coupling of aldehydes with N-hydroxyphthalimide (NHP) esters[29,30]. Laterly, pyridinium salts[31], Togni reagents[32–34], and activated perfluoroalkyl reagents[35] were used as the alkyl radical precursors for the NHC-catalyzed redox reactions. In addition, the generation of radical intermediates via photoredox has also been explored in NHC catalysis (Fig. 1b)[36–40]. Our group reported the merging of NHC catalysis and photoredox catalysis for γ- and ε-alkylation with electron-deficient alkyl halides[41]. Scheidt[42,43] and Studer[44] developed the dual NHC and photoredox catalysis using benzyl Hantzsch esters or the Langlois reagents as alkyl radical precursor, respectively. However, compared to these active alkyl radical precursors, the utility of simple but unactivated alkyl halides remains underexplored due to their low reduction potential[45]. In the last decade, the combination of NHC with TM catalysis has made great progress to promote reaction otherwise impossible (Fig. 1c)[46]. While most

[1]Beijing National Laboratory for Molecular Sciences, CAS Key Laboratory of Molecular Recognition and Function, CAS Research/Education Center for Excellence in Molecular Sciences, Institute of Chemistry, Chinese Academy of Sciences, 100190 Beijing, China. [2]University of Chinese Academy of Sciences, 100049 Beijing, China. [3]These authors contributed equally: You-Feng Han, Ying Huang. ✉e-mail: zhangchunlin@iccas.ac.cn; songye@iccas.ac.cn

**a** Redox NHC Catalysis

**d** Photoredox Cooperative NHC/Pd Catalysis (this work)

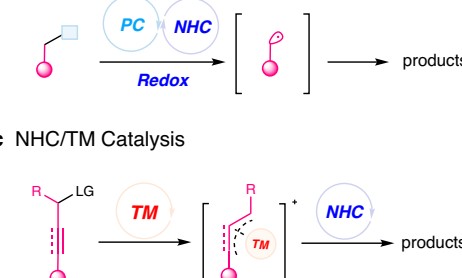

**b** Photoredox/NHC Catalysis

**Reaction design**: Photoredox NHC/Pd-Catalysed Alkylacylation of Alkenes

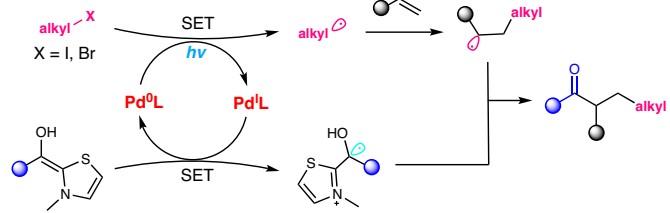

**c** NHC/TM Catalysis

\* Simple alkenes and unactivated alkyl halides

\* Three-component  \* Broad functional group compatibility

**Fig. 1 | Redox NHC and cooperative NHC/TM catalysis. a** Redox NHC-catalyzed radical reactions. **b** The merging of photoredox and NHC catalysis. **c** The combination of NHC with TM catalysis. **d** Our strategy: photoredox cooperative NHC/ palladium-catalyzed alkylacylation of alkenes. TM transition metal, LG leaving group, X halide, SET single-electron transfer.

of the reactions involve the TM-allylic/propargyl intermediate[47–55], the reaction of simple and less active alkene meets challenge[56].

Here, we develop a photoredox cooperative NHC/Pd-catalyzed alkylacylation of simple alkenes with aldehydes and unactivated alkyl halides (Fig. 1d). This reaction involves the coupling of ketyl radical from aldyes[57] and alkyl radicals from halides[58–61], delivering a range of ketones in good to high yields.

## Results

### Condition optimization of the alkylacylation

Initially, the model alkylacylation of 2-vinylnaphthalene **1a** with 2-pyridinecarboxaldehyde **2a** and commercially available trimethyl(iodomethyl)silane **3a** was explored under photoredox cooperative NHC/Pd catalysis (Table 1). We were happy to find the desired three-component coupling product **4** was isolated in 78% yield without Heck-type byproduct when the reaction was carried out in the presence of 10 mol% of Pd(OAc)$_2$ with $^t$Bu-Xantphos **L1** as the ligand, 20 mol% thioazolium preNHC **N1** with Cs$_2$CO$_3$ as the base and 20 mol% 4-methylpyridin-2-ol **A1** as the additive[62] in 1,4-dioxane under blue LED irradiation (entry 1). Control experiments showed that the yield was somewhat decreased without additive or use of additive **A2** instead of **A1** (entries 2 and 3). The use of ligand Cy-Xantphos **L2**, Xantphos, DPEphos or *rac*-BINAP instead of **L1** resulted in trace product (entries 4–5). Using THF or toluene as the solvent showed no strong influence on this reaction (entry 6). Other palladium catalysts such as Pd(PPh$_3$)$_2$Cl$_2$, Pd(PPh$_3$)$_4$ and Pd$_2$(dba)$_3$ were far less effective (entry 7). The screening of preNHC catalysts revealed that thioazolium preNHC **N2** and **N3** resulted in some loss of the yield (entry 8), while triazolium preNHC **N4** did not work, possibly due to the shorter lifespan of the ketyl radical from triazolium NHC than thioazolium[57] (entry 9). Screening of bases showed that DBU and K$_2$CO$_3$ performed much better than Na$_2$CO$_3$ possibly due to its lower basicity (entries 10–11). Lowering the loading of preNHC **N1** to 10 mol% resulted in decreased yield (entry 12). Finally, the reaction did not work in the absence of light, preNHC, Pd(OAc)$_2$, or Ligand **L1** (entry 13).

### Substrate scope

With the optimized reaction conditions in hand, the generality of the photoredox cooperative NHC/palladium-catalyzed alkylacylation reaction was then investigated (Fig. 2). It was found that all styrenes with electron-rich and electron-deficient substituents at the *para*

positions of the phenyl moiety are capable partners for this transformation, providing the desired ketones in moderate to good yields (**5–16**). Functional groups such as aldehyde, nitro, and alkynyl were also compatible (**17–19**). Both 1-vinylnaphthalene and styrenes with *meta*- or *ortho*-substituents worked well (**20–24**). Pentafluorostyrene and those with heteroaryl such as 2-pyridyl, 2-thienyl and 2-benzofuryl group were also amenable, affording the corresponding ketones in moderate to good yields (**25–28**). In addition, the reaction of 1-methyl-1-phenylethylene went smoothly to give ketone **29** with quaternary carbon in 71% yield. However, 1*H*-indene (internal alkene), allylbenzene (aliphatic alkene), and butyl methacrylate (Michael acceptor) were unreactive under current conditions.

The scope of aldehydes was then explored. Heteroaromatic aldehydes, such as furfural, thiazole-4-carboxaldehyde, 2-methylthiazole-4-carboxaldehyde, 2-pyridinecarboxaldehydes and 4-pyridinecarboxaldehyde, showed good reactivity under standard conditions (**30–36**). Benzaldehyde and those with *para*- and *meta*-substituents (R = 4-F, -Cl, -Br, -CF$_3$, -CN and 3-Cl) all worked for the alkylacylation reaction, giving the corresponding α-aromatic ketones in good yields (**37–43**). It is noteworthy that the acidic hydroxyl group in salicyladehydes showed no harm to the reaction, giving the expected ketones **44–46** in good yields.

Variation of the halides showed that all primary haloalkanes with less or more steric alkyl group reacted efficiently to furnish the coupling products **47–50** in good to high yields. Both secondary and tertiary haloalkanes worked well for the reaction, although two equivalent alkenes were employed to circumvent possible direct coupling of aldehydes with the more stable alkyl radicals (**51–58**). The additional chlorine in bromoalkanes or idoalkanes showed no negative effect for the reaction with exclusive chemical selectivity (**59–61**). The reaction of trifluorohaloalkanes, such as 1,1,1-trifluoro-2-iodoethane and 1,1,1-trifluoro-3-iodopropane, went smoothly to give the fluorinated ketones **62–63** in high yields. The benzylic radical worked as well as the alkyl ones, affording the corresponding ketone **64** in good yield. The additional ether or amine functional group in alkyl halides were tolerable (**65–66**). Furthermore, the α-haloesters, such as bromoacetate, 2-methyl-2-bromopropionate and difluorobromoacetate, were also suitable radical precursors for the reaction (**67–69**).

Cyclopropanes are privileged structure motif in natural products and bioactive compounds and useful building blocks in organic synthesis[63]. It is interesting that cholorcyclopropyl ketone **70** was

**Table 1 | Condition optimization[a]**

| Entry | Deviations from standard conditions | Yield[b] (%) |
|---|---|---|
| 1 | None | 76 (78)[c] |
| 2 | Without **A1** | 70 |
| 3 | **A2** instead of A1 | 64 |
| 4 | **L2** instead of **L1** | <10 |
| 5 | Xantphos, DPEphos, or *rac*-BINAP instead of **L1** | trace |
| 6 | THF or Tol instead of 1,4-dioxane | 66 or 68 |
| 7 | Pd(PPh$_3$)$_2$Cl$_2$, Pd(PPh$_3$)$_4$ or Pd$_2$(dba)$_3$ instead of Pd(OAc)$_2$ | 24, 27, or 38 |
| 8 | **N2** or **N3** instead of **N1** | 60 or 49 |
| 9 | **N4** instead of **N1** | trace |
| 10 | DBU or K$_2$CO$_3$ instead of Cs$_2$CO$_3$ | 64, 58 |
| 11 | Na$_2$CO$_3$ instead of Cs$_2$CO$_3$ | trace |
| 12 | 10 mol% instead of 20 mol% **N1** | 49[c] |
| 13 | Without light, or **N1**, or Pd(OAc)$_2$, or **L1** | 0 |

[a]Reaction conditions: **1a** (0.20 mmol), **2a** (0.40 mmol), **3a** (0.3 mmol), Pd(OAc)$_2$ (10 mol%), ligand (12 mol%), preNHC (20 mol%), additive (20 mol%), base (150 mol%), 2.0 mL solvent, rt, blue LED (36 W) under N$_2$ for 48 h.
[b]The yields were determined by $^1$H NMR analysis.
[c]Isolated yields.

obtained in good yield with excellent diastereoselectivity when CHCl$_3$ was used as the radical precursor for the alkylacylation reaction (Fig. 3). It was proposed that the cyclopropyl ketone was formed from alkylacylation product **70a** by an intramolecular S$_N$2 cyclization under basic condition. More examples of the cascade alkylacylation/cyclopropanation were then examined (Fig. 4). It was found that 2-naphthaldehyde and benzaldehydes substituted with 4-F, 4-Cl, and 4-CF$_3$ all worked well to give the desired cyclopropanes in good yields with good to excellent diastereoselectivities (**70–74**). In addition, when carbon tetrachloride was used, the corresponding challenging dichlorocyclopropanes with vicinal four-substituent **75–79** were afforded in moderate to good yields.

The potential utility of the reaction was demonstrated by gram-scale experiment and further chemical transformations of the resulted ketone (Fig. 5). The α-aromatic ketone **5** was obtained in 1.20 g, 81% yield from the reaction of 5.0 mmol of styrene. Alcohol **80** was resulted in 86% yield with 15:1 dr when ketone **5** was reduced by sodium borohydride (reaction a). The addition of ethynyl Grignard reagent to ketone **5** gave the corresponding alcohol **81** in 99% yield with 25:1 dr (reaction b). The Wittig reaction of ketone **5** with methyl ylide gave the corresponding olefin **82** in 67% yield (reaction c).

## Discussion
### Mechanistic studies
A series of control experiments were carried out to investigate the possible mechanism of the photoredox cooperative NHC/Pd-catalyzed

alkylacylation reaction (Fig. 6). It was found that there was no alkylacylation product **4** detected when TEMPO was added as radical scavenger, while the adduct **83** from the alkyl radical and TEMPO was detected by HRMS (Fig. 6a). The radical clock experiment using (bromomethyl)cyclopropane gave ketone **84** with rearrangement of cyclopropylmethyl radical (Fig. 6b). The reaction with 6-bromohex-1-ene gave the ketone **85a** with the formation of cyclopentyl ring as the major product (Fig. 6c). These results strongly support the involvement of the alkyl radical as the key intermediate for the reaction. The reaction of (1-(2-phenylcyclopropyl)vinyl)benzene gave ketone **86** with ring-opening of cyclopropane, which supports the involving of benzyl radical species.

The UV−Visible absorption spectra of possible combination of the substrates and reagents were then measured (Fig. 7). It was found that there was no apparent absorption for the solution of alkene, benzaldehyde, alkyl iodide, *N*-heterocylic carbene, phosphine ligand with alkyl iodide[64], or the Breslow intermediate (NHC + benzaldehyde) in range of visible light (>400 nm), while absorption was observed in the same level for Pd/ligand, Pd/ligand/NHC and the full reaction mixture under the standard conditions. These results suggested that palladium species worked as photocatalysts for the reaction (see Supplementary Fig. 1 in SI for more details).

Then light on/off experiments were conducted (Fig. 8). These results reveal that the blue light is integral in the whole reaction process and excluded the chain-reaction process.

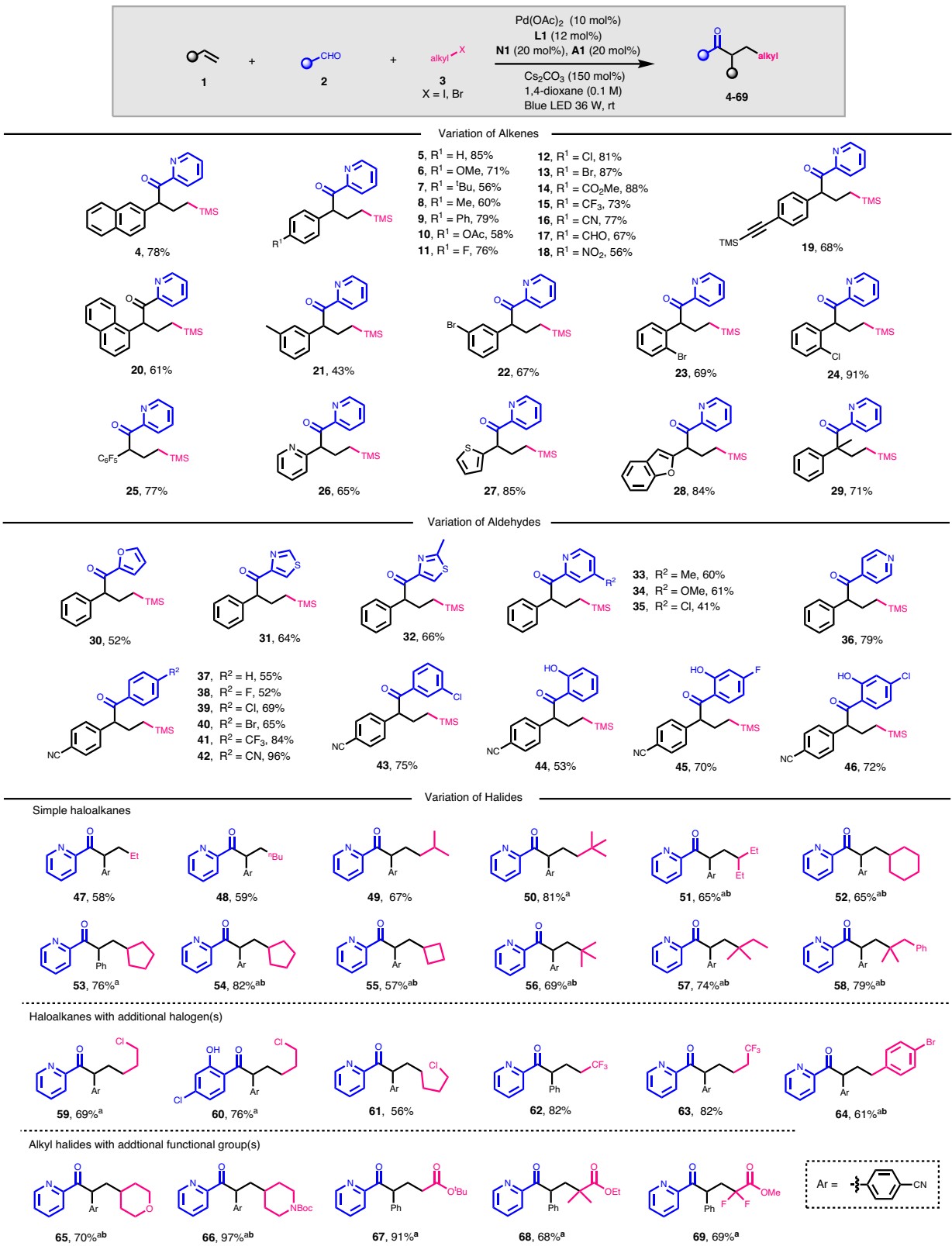

**Fig. 2 | Substrate scope for alkylacylation of alkenes.** Reaction conditions: **1** (0.20 mmol), **2** (0.40 mmol), **3** (0.3 mmol), Pd(OAc)$_2$ (10 mol%), **L1** (12 mol%), **N1** (20 mol%), **A1** (20 mol%), Cs$_2$CO$_3$ (150 mol%), 2.0 mL 1,4-dioxane, rt, blue LED (36 W) under N$_2$; X = I, unless otherwise noted. [a]X = Br; [b]**1** (0.40 mmol), **2** (0.20 mmol), and **3** (0.3 mmol) were used.

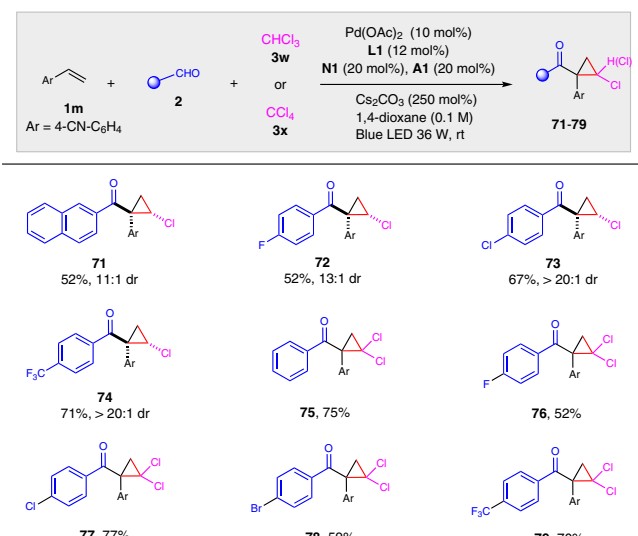

**Fig. 3 | Synthesis of cholorcyclopropyl ketone 70.** Alkylacylation using chloroform as alkyl halide gave chloorocyclopropyl ketone as the final product.

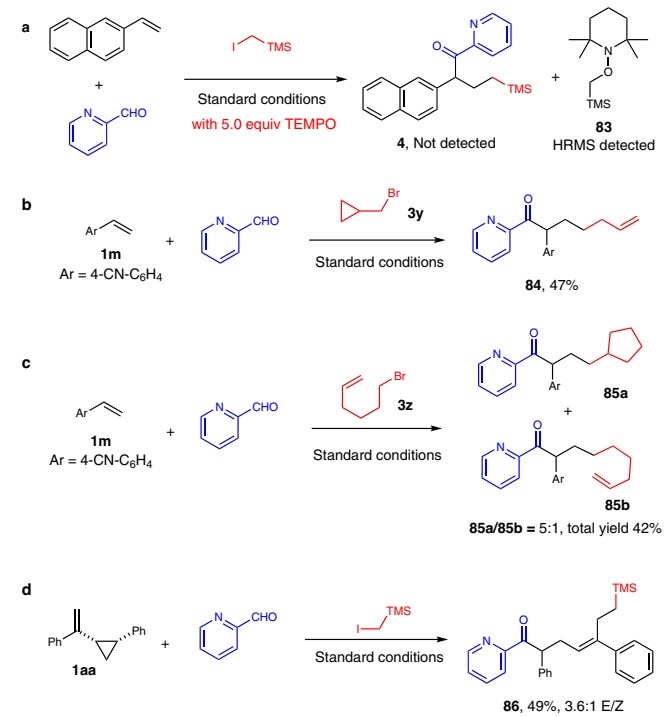

**Fig. 6 | Mechanistic study. a** Radical trapping experiment with TEMPO (2,2,6,6-tetramethylpiperidinooxy). **b** Radical clock experiment with (bromomethyl)cyclopropane. **c** Radical probe experiment with 6-bromohex-1-ene. **d** Radical clock experiment with (1-(2-phenylcyclopropyl)vinyl)benzene.

**Fig. 4 | Cascade alkylacylation/cyclopropanation.** Reaction conditions: **1m** (0.20 mmol), **2** (0.40 mmol), **3w** (2.0 mmol) or **3x** (1.0 mmol), Pd(OAc)$_2$ (10 mol%), **L1** (12 mol%), **N1** (20 mol%), **A1** (20 mol%), Cs$_2$CO$_3$ (250 mol%), 2.0 mL 1,4-dioxane, rt, blue LED (36 W) under N$_2$.

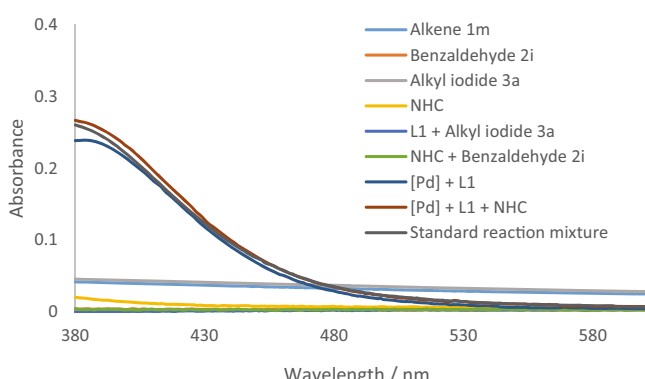

**Fig. 7 | UV–Visible absorption spectra.** The UV–Vis absorption spectra of possible combinations of the substrates and reagents.

**Fig. 5 | Gram-scale synthesis and further chemical transformations. a** NaBH$_4$, MeOH, 0 °C to rt. **b** THF, 0 °C to rt. **c** THF, 0 °C to rt.

Based on this information, the plausible catalytic cycle of the reaction was proposed as shown in Fig. 9. The Pd$^0$ complex is excited under blue light irradiation, followed by a single-electron transfer (SET) between the photoexcited Pd$^0$ complex and alkyl halides **3** to give the Pd$^I$ complex and the alkyl radical species **I**. The addition of the alkyl radical to styrenes **1** produces radical species **II**. In the meantime, the Breslow intermediate **III**, generated from aldehydes **2** under NHC catalysis, is oxidized via an SET by Pd$^I$ complex to afford ketyl radical **IV** and Pd$^0$ complex. The radical-radical coupling between **II** and **IV** gives adduct **V**, which is fragmented to release the alkylacylation products and regenerate NHC catalyst.

In summary, we reported the photoredox cooperative NHC/Pd-catalyzed alkylacylation of alkenes with simple alkyl halides. This multicomponent coupling reaction used readily available starting materials with broad functional group compatibility. A wide range of aromatic aldehydes and simple alkyl halides directly coupled with styrenes, providing synthetically useful ketones in good to high yields. In addition, a series of chloro-cyclopropanes with quaternary carbon were obtained when chloroform or carbon tetrachloride was used as the alkyl radical precursor. This method does not require exogenous photosensitizers or external reductants. Mechanistic studies suggest the involvement of the alkyl radicals from halides and the ketyl radicals from aldehydes under photoredox cooperative NHC/Pd catalysis. Further mechanistic studies and reaction development are underway in our laboratory.

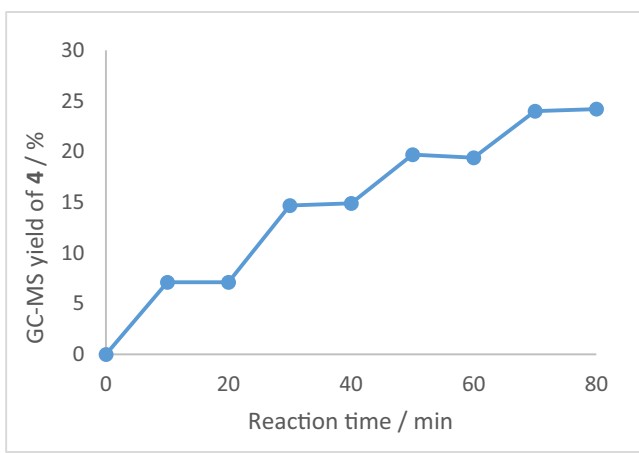

**Fig. 8 | Light on/off experiments.** The yield of ketone **4** determined by GC-MS using dodecane as an internal standard.

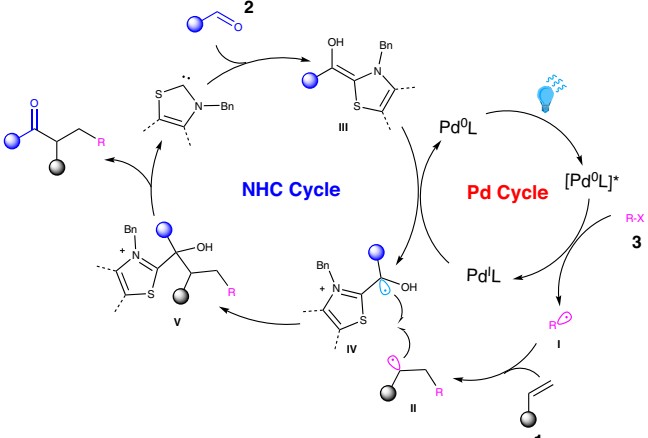

**Fig. 9 | Proposed catalytic cycle.** The plausible mechanism for the photoredox cooperative NHC/Pd catalylsed alkylacylation of alkenes.

## Methods

**General procedure for photoredox cooperative NHC/Pd-catalyzed alkylacylation of alkenes.** A 4 mL vial equipped with a stir bar was charged with preNHC N1 (10.8 mg, 0.04 mmol), Pd(OAc)2 (4.5 mg, 0.02 mmol), ligand L1 (11.9 mg, 0.024 mmol) and 1.0 mL of 1,4-dioxane. After stirring for 30 min in glove box, to the solution was added $Cs_2CO_3$ (97.8 mg, 0.3 mmol), additive A1 (4.4 mg, 0.04 mmol), alkenes 1 (0.2 mmol), aldehydes 2 (0.4 mmol), alkyl halides 3 (0.3 mmol), and 1.0 mL of 1,4-dioxane. The reaction mixture was removed from the glove box and stirred under 36 W blue LED lights at room temperature until the complete consumption of 1 (generally 48 h) by TLC analysis. The reaction mixture was filtered through a small pad of silica and eluted with EtOAc. The solution was concentrated under reduced pressure, and purified by column chromatography on silica gel to afford the desired ketones (see SI for more details on experimentation).

## Data availability

The authors declare that the data supporting the findings of this study are available within the article and its Supplementary Information file. For experimental details and compound characterization data, see Supplementary Methods. For ¹H NMR, ¹³C NMR spectra, see Supplementary Figs. 3–167. Supplementary Data 1 contains raw data for Figs. 7 and 8.

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

## Acknowledgements
Financial support from the National Natural Science Foundation of China (Nos 21831008, S.Y.; 22071253, Z.H.G.) and Beijing National Laboratory for Molecular Sciences (BNLMS-CXXM-202003, S.Y.), and the Ministry of Science and Technology of China is greatly acknowledged.

## Author contributions
Y.-F.H., Y.H., C.-L.Z., and S.Y. designed, performed, and analyzed the experiments. Y.-F.H., C.-L.Z., and S.Y. co-wrote the manuscript. Y.-F.H., Y.H., H.L., Z.-H.G., C.-L.Z., and S.Y. contributed to the discussions.

## Competing interests
The authors declare no competing interests.
