## [Peer Review File · Nature Communications]

Photoredox Cooperative N Heterocyclic Carbene/Palladium-Catalysed Alkylacylation of AlkenesREVIEWER COMMENTS

Reviewer #1 (Remarks to the Author):

The manuscript by Ye and coworkers describes a three-component alkylacylation of simple alkenes with aldehydes and unactivated alkyl halides enabled by photoredox NHC/Pd cooperative catalysis, leading to alpha-branched ketones in good yields. The catalytic assembly of functional molecules from readily available feedstocks has long been receiving worldwide interest. Both alkenes and alkyl halides are abundantly produced from industry and aldehydes are also readily available. The development of new transformations to efficiently convert these chemicals to ketones is of course important. By virtue of photoredox SET process of alkyl halides mediated with Pd(0) to generate radicals and Pd(I), integrating with NHC activation of aldehydes, the three-component alkylacylation of simple alkenes with aldehydes and unactivated alkyl halides has been established. It is not only a unique bifunctionalization of styrenes, but a value-added transformation of alkyl halides. More importantly, this work expands a new dimension of Pd/NHC cooperative catalysis to create unprecedented multicomponent reactions. In this context, this work represents a significant advance in this field and will receive a broad readership. Therefore, I would like to recommend the publication of the manuscript in Nature Communications after addressing following comments, suggestions, and questions:

- (1) It has been reported that the complex of iodide and phosphine is a good photocatalyst (see: R. Shang, Y. Fu, et al, Science 2019, 363, 1429), is such a photocatalyst species existing in the reaction system? I personally suggest to measure the UV-Visible absorption spectra of alkyl iodide and ligands to see what will happen.
- (2) A1 seems to play important role in improving the reaction performance. However, how it works is not presented in the reaction cycle (Fig. 7). The authors should comment on this issue.
- (3) The direction of arrows for Pd(0)-Pd(I) in Fig. 1d and Fig. 7 looks confusing. Basically, the Breslow intermediate undergoes SET with Pd(I) to give a radical and Pd(0), but the arrow symbol directs an opposite pathway.
- (4) In page 2, the authors claimed "After evaluation, we were happy". The results of evaluation should be reported in Supporting Information.

Reviewer #2 (Remarks to the Author):

Alkenes are ubiquitous and fundamental motifs in organic molecules. Radical difunctionalization of alkenes represents one of the most practical approaches to constructing value-added compounds in the atom- and step-economic manner. In this manuscript, Ye, Zhang, and co-workers have presented a novel photo-redox cooperative NHC/Pd-catalyzed alkylacylation of simple alkenes. This is an elegant breakthrough based on the author's continued interest in the development of NHC-catalyzed novel transformations. The transformation proceeds under extremely mild conditions with good functional group tolerance and broad scope, providing alkylacylation products and chloro-cyclopropanes. The synthetic utility of the reaction was further demonstrated by large-scale synthesis and further chemical transformations. Finally, they have carried out mechanistic studies, which support their hypothesis. Overall, the experiments are carefully performed, the manuscript is well organized, and the products are fully characterized. Hence, this reviewer recommends its publication in Nature Communications after addressing the following minor issues:

1. Reaction could be efficiently carried out by employing thioazolium preNHC including N1-N3. However, triazolium preNHC N4 seems not efficient. A further explanation should be provided.
2. Up to 75 substrates were explored, and the substrate's scope seems to be robust. How about the internal alkenes, aliphatic olefins, and alkenes with electron withdrawing groups in Michael system? A couple of examples, even failed examples, should be included.
3. The radical clock experiment strongly supports the involvement of the alkyl radical in this transformation. How about employing (1-cyclopropylvinyl)benzene as a radical clock, it may provide information concerning benzyl radical species.
4. Light on/off experiments was encouraged to get further information on the photocatalysis process.

5. The recent development on visible light-mediated NHC and photoredox co-catalyzed transformations should be cited, DOI: 10.1039/d1sc06100c; DOI: 10.1038/s41467-022-30583-2.

Point-by-Point Response

Reviewer: #1

Q1:

- It has been reported that the complex of iodide and phosphine is a good photocatalyst (see: R. Shang, Y. Fu, et al, Science 2019, 363, 1429), is such a photocatalyst species existing in the reaction system? I personally suggest to measure the UV-Visible absorption spectra of alkyl iodide and ligands to see what will happen.

Answer:

Thanks for the concern. The paper (Science 2019, 363, 1429) is cited as ref. 64. The UV-Visible absorption spectra of alkyl iodide and ligand **L1** was measured and it was found that there was no apparent absorption in range of visible light (> 400 nm). So the complex of alkyl iodide and phosphine does not work as a photocatalyst species in this work.

Q2:

- **A1** seems to play important role in improving the reaction performance. However, how it works is not presented in the reaction cycle (Fig. 7). The authors should comment on this issue.

Answer:

The yield of desired product decreased slightly to 70% from 76% without additive **A1** 4-methylpyridin-2-ol (entry 2, Table 1). The additive **A1** may works as a competitive ligand (DOI: 10.1021/jacs.9b13537), and its acidity may be helpful for the reaction.

Q3:

- The direction of arrows for Pd(0)-Pd(I) in Fig. 1d and Fig. 7 looks confusing. Basically, the Breslow intermediate undergoes SET with Pd(I) to give a radical and Pd(0), but the arrow symbol directs an opposite pathway.

Answer:

Thanks. The arrows for Pd(0)-Pd(I) in Fig. 1d have been redrawn to make it clear.

Q4:

- In page 2, the authors claimed “After evaluation, we were happy …..” . The results of evaluation should be reported in Supporting Information.

Answer:

Thanks. All results of evaluation were shown in Table 1. The text has been revised as follows: We were happy to find the desired three-component coupling product 4 was isolated in 78% yield without Heck-type byproduct when the reaction was carried out in the presence of 10 mol% of Pd(OAc)₂ with tBu-Xantphos L1 as the ligand, 20 mol% thioazolium preNHC N1 with Cs₂CO₃ as the base and 20 mol% 4-methylpyridin-2-ol A1 as the additive in 1,4-dioxane under blue LED irradiation (entry 1).

Reviewer: #2

Q1:

- Reaction could be efficiently carried out by employing thioazolium preNHC including **N1-N3**. However, triazolium preNHC **N4** seems not efficient. A further explanation should be provided.

Answer:

Thanks. The following comment has been added: The triazolium preNHC N4 did not work possibly due to shorter lifespan of the ketyl radical from triazolium NHC than thioazolium (*Angew. Chem. Int. Ed.* **2021**, *60*, 26783.).

Q2:

- Up to 75 substrates were explored, and the substrate's scope seems to be robust. How about the internal alkenes, aliphatic olefins, and alkenes with

electron withdrawing groups in Michael system? A couple of examples, even failed examples, should be included.

Answer:

Thanks. The text has been revised as followed: However, 1*H*-indene (internal alkene), allylbenzene (aliphatic alkene) and butyl methacrylate (Michael acceptor) were unreactive under current conditions.

Q3:

- The radical clock experiment strongly supports the involvement of the alkyl radical in this transformation. How about employing (1-cyclopropylvinyl)benzene as a radical clock, it may provide information concerning benzyl radical species.

Answer:

Thanks. The reaction of (1-(2-phenylcyclopropyl)vinyl)benzene gave ketone **86** with ring-opening of cyclopropylmethyl radical, which further support the involving of benzyl radical species.

Q4:

- Light on/off experiments was encouraged to get further information on the photocatalysis process.

Answer:

Thanks. Light on/off experiments were conducted. These results reveal that the blue light is integral in the whole reaction process and excluded the chain reaction process.

Q5:

- The recent development on visible light-mediated NHC and photoredox co-catalyzed transformations should be cited, DOI: 10.1039/d1sc06100c; DOI: 10.1038/s41467-022-30583-2.

Answer:

Thanks. The related references have been added as ref. 27 and 28.

REVIEWERS' COMMENTS

Reviewer #1 (Remarks to the Author):

The authors have done very careful revisions and addressed all the concerns from this reviewer. No further revision is required and the manuscript is suitable for publication.

Reviewer #2 (Remarks to the Author):

The manuscript was well revised and can be published at present form.

Point-by-point response to the reviewers

manuscript NCOMMS-22-17817A

Reviewer #1 (Remarks to the Author):

The authors have done very careful revisions and addressed all the concerns from this reviewer. No further revision is required and the manuscript is suitable for publication.

Response. Thanks for the support.

Reviewer #2 (Remarks to the Author):

The manuscript was well revised and can be published at present form.

Response. Thanks for the support.